# Two Fe-Zr-B-Cu Nanocrystalline Magnetic Alloys Produced by Mechanical Alloying Technique

Jason Daza, Wael Ben Mbarek, Lluisa Escoda, Joan Saurina and Joan-Josep Suñol *

Department of Physics, University of Girona, 17003 Girona, Spain; jason.daza@cadscrits.udg.edu (J.D.); u1930157@campus.udg.edu (W.B.M.); joan.saurina@udg.edu (J.S.)
* Correspondence: joanjosep.sunyol@udg.edu

**Abstract:** Fe-rich soft magnetic alloys are candidates for applications as magnetic sensors and actuators. Spring magnets can be obtained when these alloys are added to hard magnetic compounds. In this work, two nanocrystalline Fe-Zr-B-Cu alloys are produced by mechanical alloying, MA. The increase in boron content favours the reduction of the crystalline size. Thermal analysis (by differential scanning calorimetry) shows that, in the temperature range compressed between 450 and 650 K, wide exothermic processes take place, which are associated with the relaxation of the tensions of the alloys produced by MA. At high temperatures, a main crystallisation peak is found. A Kissinger and an isoconversional method were used to determine the apparent activation of the exothermic processes. The values are compared with those found in the scientific literature. Likewise, adapted thermogravimetry allowed for the determination of the Curie temperature. The functional response has been analysed by hysteresis loop cycles. According to the composition, the decrease of the Fe/B ratio diminishes the soft magnetic behaviour.

**Keywords:** mechanical alloying; thermal analysis; soft magnetic; Fe based

## 1. Introduction

In magnetic alloys, it is important to check the thermal stability of the magnetic phase and their soft-hard behaviour. It is also known that magnetic properties change if the alloy is amorphous or crystalline. Mechanical alloying is a production technique that is applied in the development and manufacturing of powdered Fe-rich nanocrystalline alloys [1,2]. Mechanical alloying is a powder metallurgy technology applied before sintering or consolidation [3,4]. The thermal stability of these alloys' front crystalline growth is determinant, due to the loss of soft behaviour as the crystalline size increases. A key parameter to determine this thermal stability is the apparent activation energy of crystallisation [5,6]. The MA method is one of the most preferred because of its high potential to produce metastable alloys, such as nanocrystalline and amorphous alloys.

Soft ferromagnetic alloys are characterised by low coercive field, $H_c$, high saturation magnetic flux density, $B_S$, and high permeability, μ [7]. Low coercivity and high permeability favour low core losses in applications under alternate current magnetic fields. Thus, the control of these parameters is associated with the optimisation of energy savings [8]. Regarding the saturation magnetic flux density, higher values favour application in low dimensional systems as the consequent miniaturisation [9]. Likewise, the thermal behaviour of these alloys allows for the establishment of working temperature limits; the Curie temperature marks the transition from ferromagnetic to paramagnetic and the loss of soft magnetism. Magnetic thermogravimetry has been applied to determine this limiting temperature [10]. Regarding MA, this technique favours the formation of nanocrystalline (including super saturated solid solutions or high entropy alloys) and amorphous soft magnetic alloys. The optimised selection of the milling parameters of MA improve the soft magnetic response [11]. One of the pathways to modifying the soft response is the

controlled addition of a small percentage of other elements. It has been found that the addition of non-magnetic elements, such as Cr and Nb, reduces both the magnetisation of saturation and the coercivity [12].

There are other powder techniques, such as atomisation [13]. Atomisation is also used to obtain soft magnetic particles with superior magnetic response [14–16]. In addition, studies analyse the influence of the atomisation production parameters (pressure, gas temperature and thermal conductivity, cooling rate) in the final product [17]. The atomisation process facilitates the formation of spherical particles, whereas MA particles are usually smoothed (but are not spherical). Thus, MA powders have more specific surfaces and irregular shaped particles than atomised particles of a similar radius. Gas atomisation favours the formation of the amorphous phase, however, if a high dispersion of particle sizes is produced, depending on the particle size, the structure can be amorphous or a mixture of amorphous and nanocrystalline [18]. With this technique, contamination from the MA milling tools is avoided.

In recent decades, several families of nanocrystalline soft magnetic alloys have been developed as alternatives to traditional ferrites such as Finemet [19], Nanoperm [20], Hitperm [21], or Nanomet [22]. Figure 1 shows the typical values of these nanocrystalline alloys, representing the initial permeability as a function of the saturation of the magnetic flux density. For comparison, information about traditional ferrites, sendust, permalloy, Si steels, Fe-based amorphous, and Co-based amorphous is also provided in this graph. High values of saturation flux density favour the use of these materials in technological applications in which miniaturisation is desirable. Contrastingly, a high magnetic permeability usually favours energy savings.

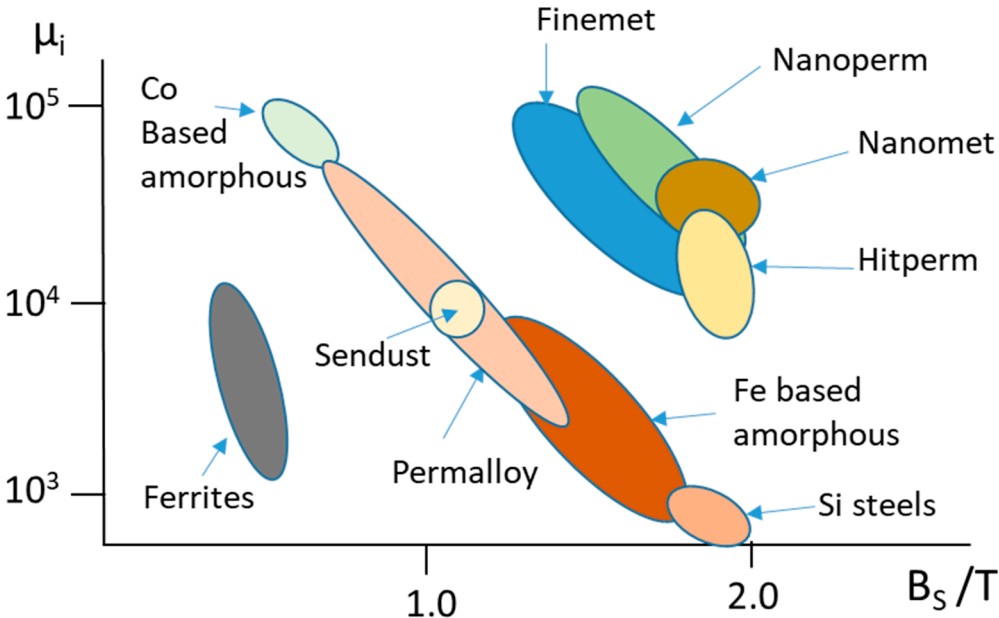

**Figure 1.** Permeability versus magnetic flux density of several families of soft magnetic alloys.

In this work, two nanocrystalline Fe-Zr-B-Cu Nanoperm-type alloys have been produced by mechanical alloying. The thermal stability has been determined through thermal analysis by checking the apparent activation energy of the main crystalline growth process and the Curie temperature.

The main difference between the two Fe-Zr-B-Cu alloys produced is the relative content of Fe and B. In one the ratio is 85/8 and in the other it is 80/13. As iron is the element that contributes magnetism to the alloy, it is expected that the magnetisation of the sample with a ratio of 85/8 is greater than that of the ratio of 80/13. However, it is known that the reduction of the nanocrystalline size favours the magnetic response (usually by decreasing coercivity) and that the addition of boron in the Fe base alloys favours either the

formation of an amorphous phase or the formation of a nanocrystalline phase with smaller nanocrystals. Likewise, the amorphous induced phase formation is usually associated with a loss of the magnetisation of saturation [23], which is an undesired effect for soft magnetic behaviour. It is therefore a question of ascertaining in two specific compositions whether one effect can counteract, at least partially, the other or not. Furthermore, it is a question of verifying the thermal stability of both alloys, since the crystalline growth (provoked on heating the samples) usually entails the loss of soft magnetic behaviour. The percentage of copper and zirconium is the same in both alloys, therefore its effect on the different thermal and thermomagnetic responses, or on the microstructure, can be considered lesser due to the relative content of Fe and Nb. Looking at its role, the Nb being an element of high atomic size, it will normally be located on the neighbouring crystalline grains, hindering the crystalline growth, whereas the well-dispersed copper atoms favour the formation of multiple nanocrystals and a higher density of nanocrystals, making the formation of large-sized crystals difficult.

## 2. Materials and Methods

In this work, two Fe-rich soft magnetic alloys were produced by mechanical alloying (MA). The initial compositions were $Fe_{85}Zr_6B_8Cu_1$ (at.%) and $Fe_{80}Zr_5B_{13}Cu_1$ (at.%); both Nanoperm-type. These alloys are labelled as A and B, respectively. The precursors were elemental Fe (6–8 μm, 99.9% purity), Zr (5 μm, 98% purity), B (20 μm, 99.7% purity), and Cu (45 μm, 99% purity) powders. The MA was performed in a planetary ball mill device (Fritsch P7 model). The MA process was achieved up to 50 h, under argon atmosphere, with hardened steel vials and balls as MA media, a ball-to-powder weight ratio (BPR) of about 10:1, and a rotation speed of 700 r.p.m. The Ar atmosphere was undertaken in a cycling process (first vacuum near $10^{-5}$ atm., second Ar addition to 1.1 atm.) performed three times. To prevent an excess in the internal vial temperature, the MA was performed in cycles (10 min on followed by 5 min off). Thus, a period of 75 h corresponds to 50 h of MA. The extractions were performed after waiting for the vials to cool down to prevent high surface oxidation of the metallic particles when vials were opened. The experimental conditions were chosen to optimise the alloys' production by the mechanical alloying technique. In the preparation of soft magnetic samples, a key parameter is to prevent excessive oxidation of the metallic particles because the formation of oxides usually reduces the magnetic response, and the oxide layer hinders the interaction between the magnetic domains and the associated exchange coupling when a bulk specimen is built.

The particles' powder morphology and distribution size were checked by scanning electron microscopy (SEM) in a DSM960A Zeiss apparatus (Zeiss, Jena, Germany). The final composition of the alloys was checked by inductive coupled plasma (ICP) in a Liberty-RL ICP Varian equipment. The nanocrystalline state (bcc Fe-rich phase) was confirmed by X-Ray diffraction (XRD) patterns, collected using D-500 Siemens (Bruker, Billerica, MA, USA) equipment with CuK$\alpha$ radiation ($\lambda$ = 0.15406 nm). The thermal stability of the mechanically alloyed powders was studied by differential scanning calorimetry (DSC) using a LabSys Evo 1600 °C apparatus (Setaram, Caliure-et-Cuire, France). The DSC curves were measured in the temperature range of 350−950 K at different heating rates: 5, 10, 20, 30, and 40 K/min under argon flow (20 mL/min). The thermogravimetry measurements were performed in a TGA Star$^e$ Mettler Toledo model under Ar atmosphere at a heating rate of 10 K/min. The magnetic parameters, such as intrinsic coercivity, *Hc*, saturation magnetisation, *Ms*, and saturation to remanence ratio, *Mr/Ms*, were determined by analysing the magnetic hysteresis loops collected in a Lakeshore 7404 vibrating sample magnetometer (VSM) at room temperature, under an applied magnetic field of 15 kOe.

## 3. Results

### 3.1. Morphology and Structure

The alloys were produced in powder form. Figure 2 shows two micrographs of alloys A and B milled for 50 h. The rounded shape of the particles, with smooth contours and

a micrometric size is verified. Moreover, a relatively wide distribution in particle sizes was found. In order to check the particles' size and distribution, the particle size of five micrographs of each sample were measured. The results are shown in Figure 3. The distribution does not correspond to a typical distribution function, either symmetric or asymmetric. Therefore, to compare both distributions, it was decided that the calculation of the median particle size would be performed, as well as calculating the values of the particle size of the first and last 10%. The calculated values were as follows: the first 10% of particles' sizes were 8.1 μm and 4.8 μm, the median particle sizes were 16.9 μm and 21.6 μm, and the last 10% of particles' sizes were 30.7 μm and 40.6 μm, respectively. As is observable, the corresponding values for the distribution of sample B were higher than those for sample A.

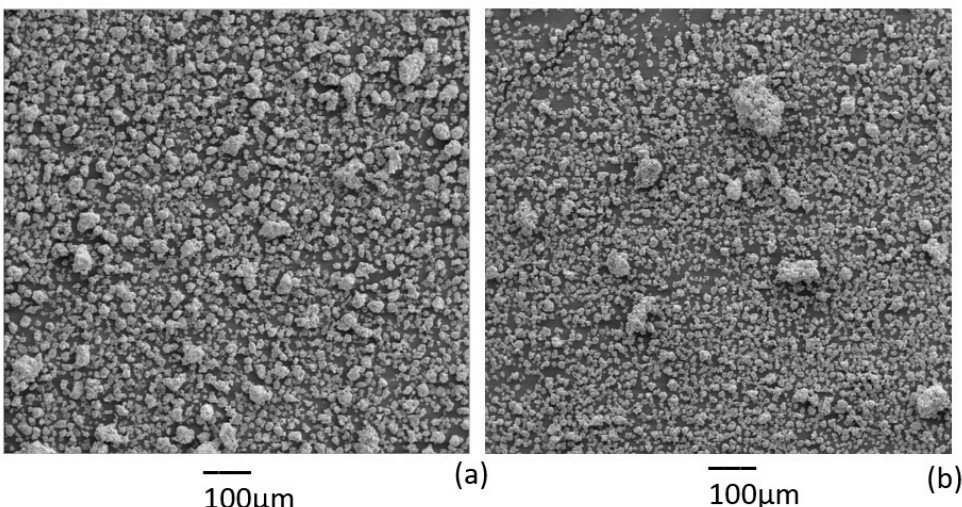

**Figure 2.** SEM micrographs of alloys A (**a**) and B (**b**) milled for 50 h.

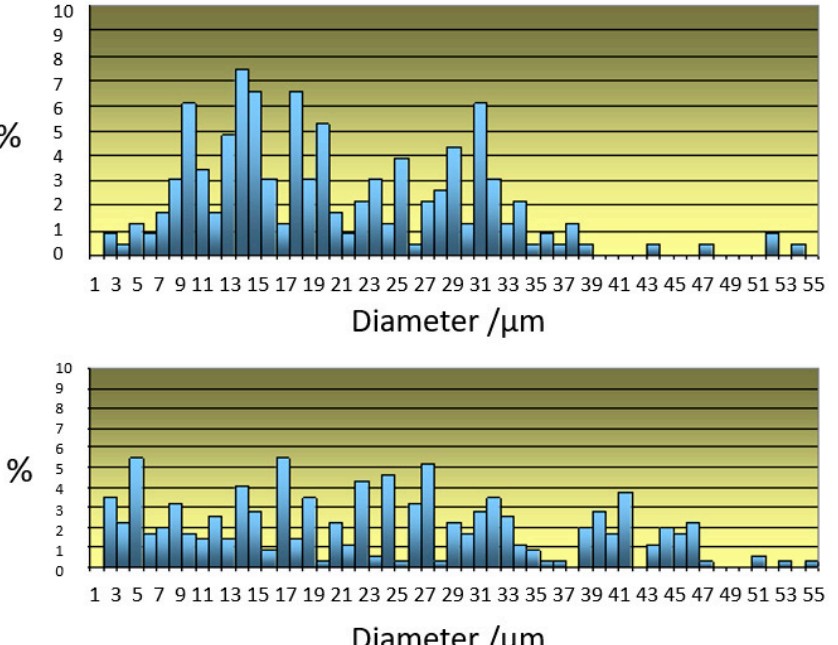

**Figure 3.** Particle size distribution of alloys A (up) and B (down) milled for 50 h.

Alternative methods for determining particle size distribution are sieve screening or laser diffraction. In this study, measurements made by sieving (three sieves and therefore four values) are consistent with those determined from microscopic observation.

The contamination was measured by ICP. The mechanical alloying process favours contamination from the MA tools. Likewise, the high surface/volume ratio of the particles induces surface oxidation. Nevertheless, the results show only slight contamination from the MA tools (Fe from the container and walls) and oxygen in both alloys after 50 h of MA. Similar results were previously reported [24]. ICP results show that the Fe content increased over the nominal composition and was $1.7 \pm 0.3$, and $1.6 \pm 0.4$ at.% for alloys A and B, respectively. Similarly, the oxygen content was $2.1 \pm 0.6$ and $1.9 \pm 0.5$ at.% for alloys A and B, respectively; it is probable that a decrease in the iron percentage is associated with a reduction in the oxygen level.

Regarding microstructural analysis, XRD patterns (Figure 4) confirm the formation of the cubic bcc structure with nanocrystalline size. The analysis applying the Williamson–Hall method allows for the determination of both the crystalline size and the microstrain. For alloy A, the crystalline size is $26 \pm 2$ nm and the microstrain $0.58 \pm 0.06\%$; whereas, for alloy B, the values are $15 \pm 2$ nm (crystalline size) and $0.61 \pm 0.08\%$ (microstrain). Thus, the increase in boron content favours, as expected, the reduction of the crystalline size, whereas the microstrain (associated with the crystallography defects) is very similar for both samples.

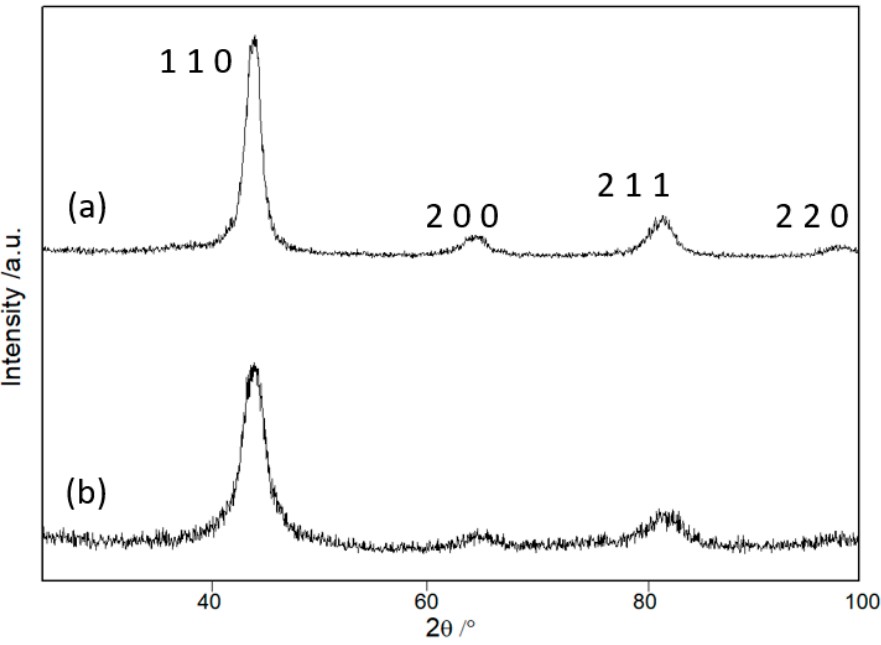

**Figure 4.** X-Ray diffraction pattern of alloys A (**a**) and B (**b**) milled for 50 h, indicating the Miller indexes of the bcc structure.

### 3.2. Thermal Analysis

Figure 5 shows the DSC curves of the two alloys milled for 50 h (heating rate: 10 K/min). The general shape of the curves is similar in both alloys. Around 400 K, a large exothermic process begins. This process is typically found in samples produced by mechanical alloying [25]. Its origin is the microstructural relaxation of the high density of crystallographic defects induced during MA grinding. The existence of small exothermic processes of the same character at higher temperatures could be an indicator of a non-completely homogeneous sample. The weak exothermic process around 750 or 780 K in both samples is typical of a partial recrystallisation of the material. The main exothermic process (in the form of a peak, with temperatures around 850 and 880 K in each sample) is due to the crystalline growth of the Fe-rich bcc phase. The development of high-performance soft magnetic alloys is associated with amorphous and low-crystalline-size nanocrystalline alloys. It is necessary that the crystalline size remains smaller than 10 nm. In this case, by applying the random anisotropy model, Hc depends on $D^6$ because the

domain wall effect diminishes, and each grain behaves as a single domain. Thus, crystalline growth should be avoided, and the crystallisation temperature is a limiting temperature for the application of these alloys [26]. It should be remarked that as the Fe/B ratio lowers, the main crystallisation peak is shifted to higher temperatures (about 20 K). Thus, partial substitution of Fe by B increases the thermal stability of the original nanocrystalline phase produced in the mechanical alloying process.

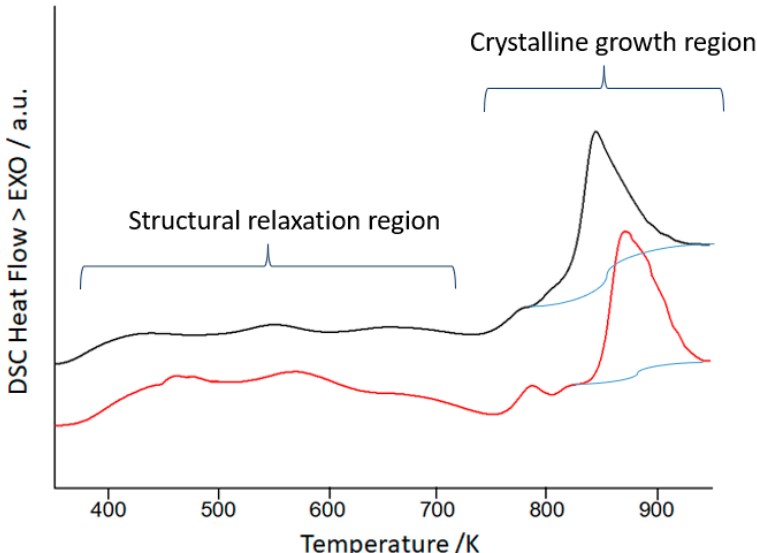

**Figure 5.** DSC scans at 10 K/min of the as milled alloys A (black) and B (red), indicating the structural relaxation and crystalline growth intervals, as well as the baseline of the main crystallisation process in both samples (blue lines).

One of the most characteristic parameters for the characterisation of crystallisation process is its activation energy. The apparent activation energy of the main crystallisation process was calculated by applying the Kissinger method [27], the typical representation of which (for both analysed alloys) is shown in Figure 6. This method is based on the determination of the peak temperature of the crystallisation process in the experiments carried out at different heating rates. The values are 282 $\pm$ 26 and 299 $\pm$ 25 kJ/mol for alloys A and B, respectively. These values are consistent with those of the crystallisation of the bcc Fe-rich phase.

The calculated energies of this work are compared with some of those found in the scientific literature (applying Kissinger method) in Table 1.

**Table 1.** Apparent activation energy of the crystallisation processes (Fe-rich bcc phase).

| Composition at.% | Activation Energy kJ mol$^{-1}$ | Initial Structure | Reference |
|---|---|---|---|
| $Fe_{83}P_{16}Cu_1$ | 238 | Amorphous | [28] |
| $Fe_{68}Nb_6B_{23}Mo_3$ | 310 | Amorphous | [29] |
| $Fe_{80}Si_{20}$ | 245 | Amorphous | [30] |
| $Fe_{83}P_{16}Cu_1$ | 219 | Nanocrystalline | [31] |
| $Fe_{83}P_{14.5}Cu_1Al_{1.5}$ | 238 | Nanocrystalline | [31] |
| $Fe_{78}Si_{11}B_9$ | 370 | Amorphous | [32] |
| $Fe_{73.5}Cu_1B_7Si_{15.5}Nb_3$ | 295 | Nanocrystalline | [32] |
| Fe (99.9% purity) | 224 | Nanocrystalline | [33] |
| $Fe_{71}Si_{16}B_9Cu_1Nb_3$ | 341 | Amorphous | [34] |
| $Fe_{85}Zr_6B_8Cu_1$ | 282 | Nanocrystalline | This Work |
| $Fe_{80}Zr_5B_{13}Cu_1$ | 299 | Nanocrystalline | This Work |

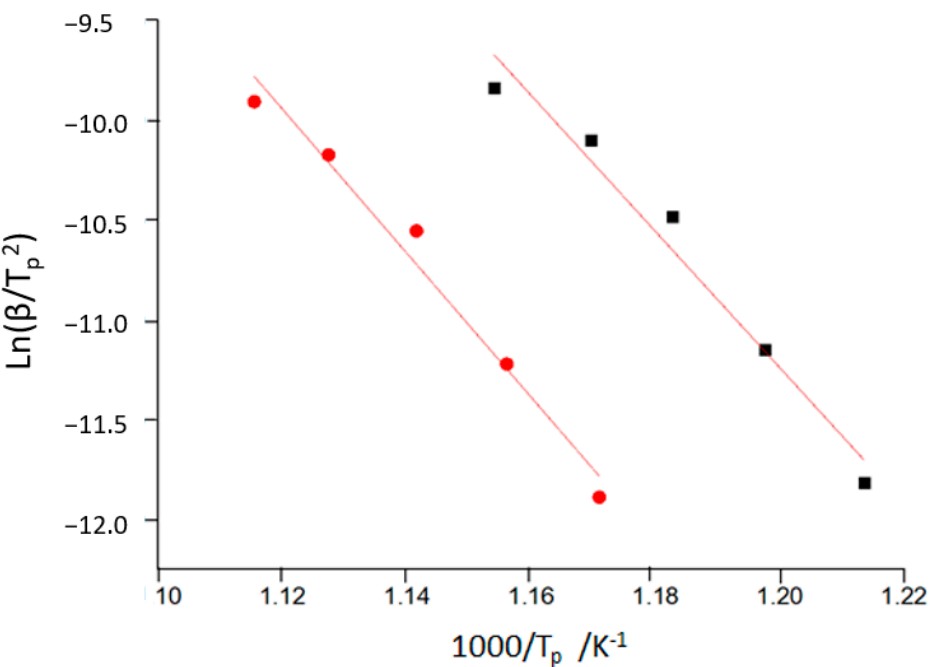

**Figure 6.** Kissinger plot of the milled alloys A (black) and B (red).

This work's values are in the same relatively wide range [28–34]. The shift in the activation energies can be related to different composition and microstructure (including crystallographic defects). This dispersion is also found in research on alloys of similar composition. Liu et al. [35] found values ranged between 138 and 356 kJ/mol in Fe-Ni-Zr-B alloys produced by mechanical alloying. The addition of Cu facilitates the reduction of (a) the crystallisation temperature and (b) the activation energy [36]. In Fe-based alloys, with a minor addition of Cu, values ranging from 177 to 233 kJ/mol were calculated [37]. Thus, the identification of the process of crystallisation or as crystalline growth is usually performed while taking into account the room temperature nanocrystalline state [26,34]. Likewise, the formation of borides is found at higher temperatures when compared with those of bcc Fe-rich crystallisation [38,39].

Likewise, there are other linear methods (normally based on the determination of the peak temperature). However, the activation energy was similar to those calculated by Kissinger, and the differences are based more on the linear relationships established between the parameters than on real differences in the crystallisation process [29]. For example, representing $\ln (\beta/T_p^2)$ is not the same as representing $\ln (\beta/T_p)$ (in both cases as a function of the inverse of the peak temperature).

The scientific literature on the activation energy of crystallisation processes shows methods in which two energies are determined: that of crystal nucleation and growth [40], the second having a higher value. This approximation is not applicable when there is only crystal growth (for example, in initially nanocrystalline alloys). In recent decades, a set of methods based on the calculation of the apparent activation energy has been extended to different transformation/crystallisation fractions. These methods are defined as isoconversional [41,42]. Figure 7 shows the calculated values (for the main exothermic process) at fractions transformed from 0.1 to 0.9 [42]. A fairly stable value is found, except for high fractions where a slight decrease in activation energy is detected. In the zone of low-medium transformed fraction, the set of calculated values is similar to that determined by applying the Kissinger method, around 288 and 298 kJ/mol for alloys A and B, respectively. For nanocrystalline alloys it is normal for this value to be stable, since the crystal nucleation process is negligible [43]. Thus, the activation energy corresponds to the crystal growth mechanism. In nanocrystalline alloys with similar composition, the same phenomenon was detected [25–34]. At high transformed fractions, the degree of transformation slows

down and the local activation energy decreases [44], probably due to the higher influence of the diffusion as main mechanism. There is probably an impingement between the crystal grains. One of the aspects to highlight is that the isoconversional method applied here is not associated with any kinetic model. Therefore, it allows us to parameterise the evolution of the activation energy as a function of the transformed fraction, neglecting the effects of a complex and variable kinetics during the transformation (nucleation, growth, impingement). On the other hand, linear methods are based on concrete hypotheses. For example, the Kissinger equation (used in the analysis of the Figure 6) was obtained assuming: (a) that the fraction transformed at the peak temperature is the same in all experiments, regardless of the heating rate, and (b) the transformation rate (crystal growth in our case) is maximum at the peak temperature.

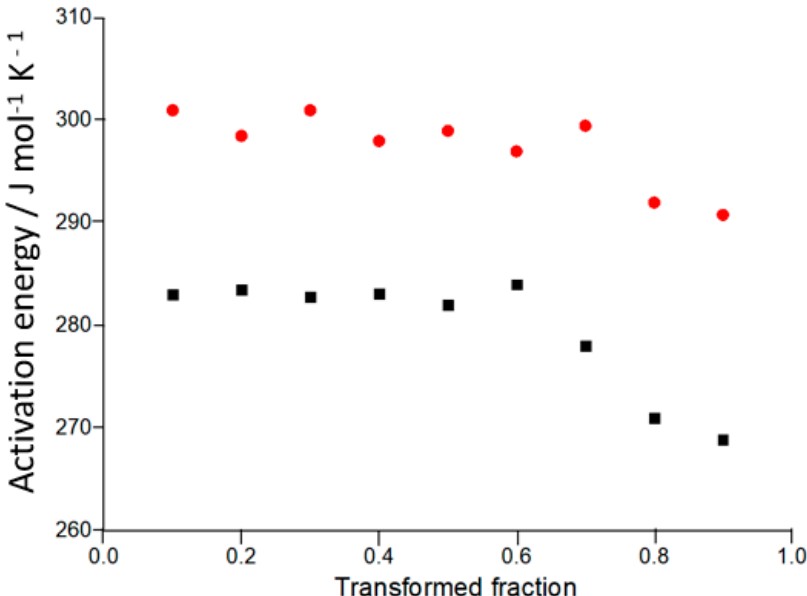

**Figure 7.** The activation energy as a function of the transformed fraction of the milled alloys A (black) and B (red).

With respect to the thermal stability of both samples after 50 h of grinding, sample B (which has a higher boron content and therefore a lower Fe/B ratio) is the one with the best thermal stability against the main crystallisation process. In general, greater thermal stability is associated with two aspects: (a) a higher transformation temperature and (b) a higher activation energy. In this study, both parameters are better in alloy B. From a technological point of view, what is desirable is a greater thermal stability of the nanocrystalline phase, since crystal growth causes an increase in coercivity and the magnetically soft response.

### 3.3. Thermomagnetic Analysis

Regarding the magnetic behaviour, one of the most characteristic and fundamental thermomagnetic parameters of magnetically soft alloys is the Curie temperature. This is the temperature that marks the maximum value of applicability (magnetic) of the alloy, since it defines the change from a ferromagnetic to a paramagnetic behaviour. The Curie temperature can be determined from the DSC curves [45]. However, in the case of overlapping processes (as is often the case with alloys produced by mechanical alloying) their determination can be complex [46]. Therefore, on many occasions this temperature is determined directly by magnetic measurements (of variation of magnetisation as a function of temperature). Another alternative, based on thermal analysis techniques, is magnetic thermogravimetry. By adding a small external magnet near the sample zone and performing the thermogravimetry experiment, an apparent variation of the sample mass during

the transition from ferromagnetism to paramagnetism is detected. This method, previously applied [47], was used in the present study. Figure 8 shows the thermogravimetric curves recorded. A variation of the apparent mass of the samples around $937 \pm 1$ K and $971 \pm 1$ K for alloys A and B, respectively, was found. These temperatures are typical of ferromagnetic alloys with a high iron content.

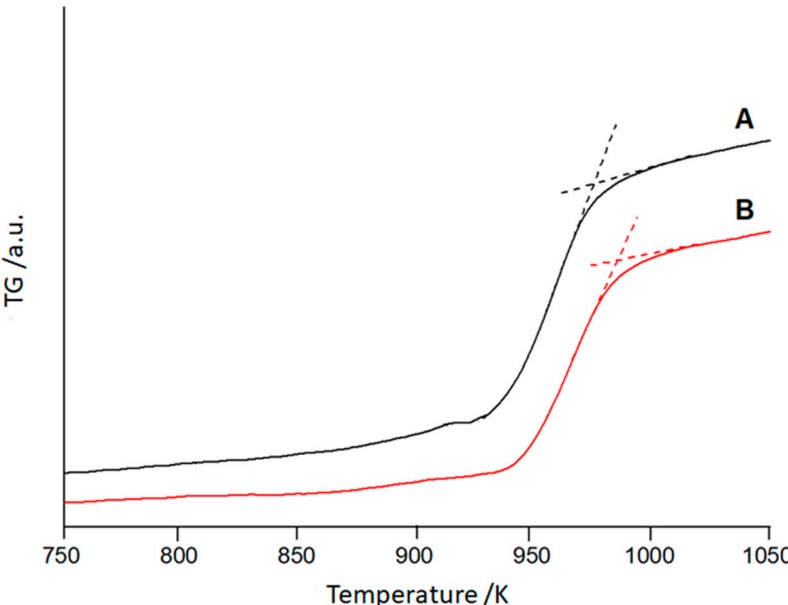

**Figure 8.** Magnetic thermogravimetry curves of alloys the as milled alloys A (black) and B (red).

The modification made to the thermogravimeter would also allow us to determine the magnetisation of the sample, although for this it would be necessary to always fix the external magnet in the same position and perform a calibration with several magnetic standards (previously analysed with another magnetic measurement technique, such as vibrating sample magnetometry). Consequently, it is better to determine the magnetisation of saturation in magnetic devices (such as a vibrating sample magnetometer). However, in consecutive measurements it is always possible to detect in which samples the change in magnetisation is greater after the transformation associated to the Curie temperature. To make the comparison, it is necessary to normalise taking into account the mass of each sample used in the experiment. In our case, it is detected that the jump is slightly lower in sample A, indicating a somewhat higher initial magnetisation. This result must be confirmed with a magnetic measurement (e.g., by magnetic hysteresis cycles).

As the interest of these alloys is based on their magnetic response, the hysteresis cycles of both alloys (after 50 h of MA) at room temperature were obtained. Figure 9 shows both magnetic cycles (magnetisation M as a function of external magnetic field H). Its analysis allows us to determine the parameters that define its soft magnetic response. The determined values are shown in Table 2. This shows magnetisation of saturation $M_s$ 146 and 139 $A \cdot m^2 \cdot kg^{-1}$, coercivity $H_C$ $12.4 \cdot 10^{-4}$ T and $10.6\ 10^{-4}$ T, and remanence 0.60 and 0.71 $A \cdot m^2 \cdot kg^{-1}$ for alloys A and B, respectively. Thus, the sample with lower nanocrystalline size has lower coercivity.

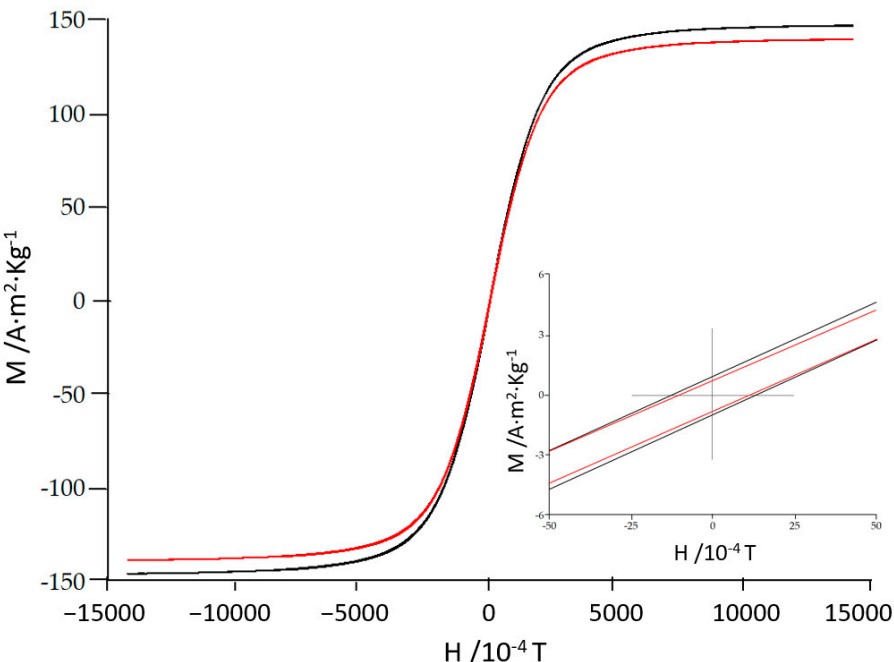

**Figure 9.** Magnetic hysteresis loops at room temperature of the milled alloys A (black) and B (red). The inset corresponds to the (0,0) region.

**Table 2.** Magnetic parameters derived from hysteresis cycles at room temperature.

| Alloy | $H_c$ $10^{-4}$ T | $M_s$ A·m$^2$·kg$^{-1}$ | $M_r$ A·m$^2$·kg$^{-1}$ | $M_r/M_s$ $10^{-3}$ |
|---|---|---|---|---|
| $Fe_{85}Zr_6B_8Cu_1$ | 12.4 | 146 | 0.60 | 4 |
| $Fe_{80}Zr_6B_{13}Cu_1$ | 10.6 | 139 | 0.71 | 5 |

It was found that the magnetisation of saturation has a slightly higher value in sample A. This result is consistent with that detected by magnetic thermogravimetry (Figure 8), where sample A shows a slightly higher variation than that of sample B at the Curie temperature (transition from ferromagnetism to paramagnetism).

The main influence will be related to the Fe atoms' local environment and to Fe-Fe interatomic distance due to the magnetic behaviour of iron. We expected to detect a diminution in the magnetisation of saturation as the Fe/B ratio decreased. Likewise, it is known that the magnetic properties depend strongly on the microstructure evolution, crystalline size, internal stress, particle shape anisotropy, magnetic anisotropy, and magnetostriction of the materials [48]. In our work, higher B content decreases crystalline size and this effect counteracts, partially (favouring soft behaviour), the effect of the reduction of the magnetic element, Fe. However, it is a minor effect on the content of the magnetic element, iron. In both alloys, when changing the percentage of iron from 85 (sample A) to 80 (sample B), a decrease in the magnetic response (saturation magnetisation) of around 6% would be expected. The decrease detected is somewhat smaller, but it is 5%. In relation to this, it should be determined that, from a magnetic point of view, in the two alloys studied the reduction of nanocrystalline size does not counteract the effect of the iron reduction. This phenomenon was to be expected, as the effect on crystal size is considered more important in coercivity. Regarding ulterior compacting to produce bulk samples, the Fe-B bonds dominated the structural compression [23].

The other values of magnetic properties are similar in both samples. The relatively low value of the squareness ratio is typical of alloys obtained in powder form by mechanical alloying [6].

Thus, we can conclude that the alloys produced are magnetically soft at room temperature and that thermal analysis is useful to determine the thermal stability (front crystallisation and front magnetic transition) of this magnetic behaviour.

## 4. Conclusions

Two ferromagnetic nanocrystalline alloys of the Fe-Zr-B system were produced by mechanical alloying (powder shape). Thermal analysis measurements allowed for the analysis of the thermal stability of the samples. High apparent activation energy, as well as high transition temperatures (crystallisation, Curie), are needed to prevent the loss of the soft behaviour caused by the increase of the crystalline size or the ferro- to paramagnetic transition. The differential calorimetry allowed us to determine (using Kissinger and isoconversional methods) the apparent energy of activation of the main process of crystallisation. The lowest value, 288 kJ/mol, corresponded to the alloy with higher iron content. Regarding the transition temperatures, both the peak crystallisation temperature and the Curie temperature are higher in the alloy with lower Fe/B ratio.

Regarding the magnetic response, the characteristic parameters are similar in both alloys, the magnetisation of saturation being only slightly higher (about 5%) in the alloy with more iron content due to the Fe-Fe magnetic moment atomic interactions.

From a technological point of view, high saturation magnetisation and low coercivity are necessary for the feasibility of the applications of these alloys in devices and systems that must present a soft magnetic behaviour. It is reported that although the saturation magnetisation is slightly lower in the alloy with a lower Fe/B ratio, the decrease in coercivity with increasing boron content (lower Fe/B ratio, alloy labelled as B) is relatively significant due to the smaller size of the nanocrystals of the low Fe/B ratio alloy. This magnetic behaviour, combined with a high thermal stability front crystalline growth (and with the associated loss of the soft magnetic behaviour), of alloy B indicates that the reduction in Fe content does not always provoke a significant decrease in the applicability of this family of alloys. With respect to future outlook and the industrial application of these types of alloys, it is necessary to thoroughly study new compositions and to analyse the consolidation process of these powders and its influence on the final microstructure and magnetic response.

**Author Contributions:** Conceptualisation, J.-J.S.; methodology, J.S.; formal analysis, W.B.M.; investigation, J.D.; writing—original draft preparation, J.D. and J.-J.S.; supervision, L.E. All authors have read and agreed to the published version of the manuscript.

**Funding:** Financial support from PID2020-115215RB-C22 project.

**Institutional Review Board Statement:** Not applicable.

**Informed Consent Statement:** Not applicable.

**Data Availability Statement:** Data will be made available upon reasonable requests to the authors.

**Conflicts of Interest:** The authors declare no conflict of interest.

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
