# Peer review of "Two Fe-Zr-B-Cu Nanocrystalline Magnetic Alloys Produced by Mechanical Alloying Technique"

_technologies, doi:10.3390/technologies11030078_

Round 1

Reviewer 1 Report

The authors have used a mechanical alloying method to prepare two nanocrystalline FeZrBCu alloys. Thay also studied the thermal and magnetic behaviors of these two alloys. 

There are some similar papers discussing the characterization of soft magnetic nanocrystalline alloys  prepared by mechanical alloying. The authors should cite these publications in the Introduction section and discuss the originality/novelty of the present work. For example, what kind of scientific question is answered in the present work? What is the novely of the present work compared to past similar work?

This manuscript should be carefully revised to correct some mechanical erros. Listed below are just some examples:

1) Line 57: in in a should be replaced by in a. 

2) Figure 2: 100um should be replaced by 100 um.

3) Line 106: 50h should be replaced by 50 h.

4) Line 107: 10K/min should be replaced by 10 K/min. 

5) Line 140: what does "sane alloy system" represent?

6) Line 101-102: Looks to me the microstrain values are quite large. Please check the value of microstrain. Are they 0.58 and 0.61 or 0.58% and 0.61%?

Author Response

Comments and Suggestions for Authors

The authors have used a mechanical alloying method to prepare two nanocrystalline FeZrBCu alloys. They also studied the thermal and magnetic behaviors of these two alloys. 

There are some similar papers discussing the characterization of soft magnetic nanocrystalline alloys  prepared by mechanical alloying. The authors should cite these publications in the Introduction section and discuss the originality/novelty of the present work. For example, what kind of scientific question is answered in the present work? What is the novelty of the present work compared to past similar work?

Answer: We modify the introduction section remarking the scientific and technological interest and novelty. Furthermore, we also add more information in the manuscript (more than one thousand new words) and we introduce 13 new references.

The main difference between the two Fe-Zr-B-Cu alloys produced is the relative content of Fe and B. In one the ratio is 85/8 and in the other 80/13. As iron is the element that contributes magnetism to the alloy, it is expected that the magnetization of the sample with a ratio of 85/8 is greater than that of the ratio of 80/13. However, it is known that the reduction of the nanocrystalline size favors the magnetic response (usually by decreasing coercivity) and that the addition of boron in the Fe base alloys favors either the formation of an amorphous phase or the formation of a nanocrystalline phase with smaller nanocrystals. Likewise, the amorphous induced phase formation is usually associated to a loss of the magnetization of saturation [23], which is an undesired effect for soft magnetic behavior. It is therefore a question of ascertaining in two specific compositions whether one effect can counteract, at least partially, the other or not. Also, to verify the thermal stability of both alloys, since the crystalline growth (provoked on heating the samples) usually entails the loss of soft magnetic behavior. The percentage of copper and zirconium is the same in both alloys, therefore its effect on the different thermal and thermomagnetic response, or in the microstructure can be considered less than that due to the relative content of Fe and Nb. Looking at its role, the Nb being an element of high atomic size will normally be located on the neighboring of the crystalline grains hindering the crystalline growth whereas the well dispersed copper atoms favor the formation of multiple nanocrystals and higher density of nanocrystals also difficult the formation of large sized crystals.

Comments on the Quality of English Language

This manuscript should be carefully revised to correct some mechanical errors. Listed below are just some examples:

Answer: The revised manuscript has been checked by a English native.

  • Line 57: in in a should be replaced by in a. 

Answer: We agree. We modify.

  • Figure 2: 100um should be replaced by 100 um.

Answer: We agree. We modify.

  • Line 106: 50h should be replaced by 50 h.

Answer: We agree and replace.

  • Line 107: 10K/min should be replaced by 10 K/min.

Answer: We agree and replace. 

  • Line 140: what does "sane alloy system" represent?

Answer: It was a typo (same). We modify to clarify.

of the alloys of similar composition.

  • Line 101-102: Looks to me the microstrain values are quite large. Please check the value of microstrain. Are they 0.58 and 0.61 or 0.58% and 0.61%?

Answer: It is the %. We modify in the manuscript (all changes yellow remarked).

Reviewer 2 Report

Fe-Zr-B-Cu magnetic alloys were produced by a mechanical alloying process. Formally, the contribution is processed relatively well. The novelty of the contribution is not very high. The contribution does not bring any surprising new knowledge about magnetically soft alloys, but neither does it pay much attention to the parameters of the technology of mechanical alloying of the alloy in question. Nevertheless, it can find readers dealing with more general knowledge in the field of the preparation of powder magnetic alloys.

Notes and recommendations:

The introduction should express the reason why mechanical alloying was used. A comparison of the properties of atomised and MA nanocrystalline soft magnetic alloys is missing. I suggest extending the introduction.

Particle size distribution measurement based on SEM observation is not the most appropriate method. Laser diffraction methods or sieve screening are more suitable and relevant methods. However, I suggest calculating the median particle size as well as the value of the particle size of the first and last 10 %.

In Fig.4. and Fig.5., I suggest adding a peaks description to increase readability.

What is the accuracy of the thermogravimetric method of Curie temperature measurement?

Author Response

Comments and Suggestions for Authors

Comment: Fe-Zr-B-Cu magnetic alloys were produced by a mechanical alloying process. Formally, the contribution is processed relatively well. The novelty of the contribution is not very high. The contribution does not bring any surprising new knowledge about magnetically soft alloys, but neither does it pay much attention to the parameters of the technology of mechanical alloying of the alloy in question. Nevertheless, it can find readers dealing with more general knowledge in the field of the preparation of powder magnetic alloys.

Answer: We add more information in the materials and methods section., also about magnetic powdered alloys.

The Ar atmosphere was done in a cycling process (first vacuum near 10-5 atm., second Ar addition to 1.1 atm.) performed three times.

Thus, period of 75 h corresponds to 50 h of MA. The extractions are performed waiting vials cooling to prevent high surface oxidation of the metallic particles when vials are opened.

In the preparation of soft magnetic samples, a key parameter is to prevent an excessive oxidation of the metallic particles because the formation of oxides usually reduces the magnetic response and the oxide layer difficult the interaction between the magnetic domains and the associated exchange coupling when a bulk specimen is built.

We add also information about MA in the introduction

Regarding MA, this technique favours the formation of nanocrystalline (including super saturated solid solutions or high entropy alloys) and amorphous soft magnetic alloys. The optimized selection of the milling parameters MA improve the soft magnetic response [11]. One of the pathways to modify the soft response is the controlled addition of small percentage of other elements. It has been found that the addition of non-magnetic elements, such as Cr and Nb, reduces both the magnetization of saturation and the coercivity [12].

Notes and recommendations:

Comment: The introduction should express the reason why mechanical alloying was used. A comparison of the properties of atomised and MA nanocrystalline soft magnetic alloys is missing. I suggest extending the introduction.

Answer: We add a paragraph in the introduction (as suggested by the reviewer) and 6 new references [13] to [18].

There are other powder techniques, such as the atomization [13]. Atomization is also used to obtain soft magnetic particles with superior magnetic response [14-16]. These studies also analyse the influence of the atomization production parameters (pressure, gas temperature and thermal conductivity, cooling rate) in the final product [17]. Atomization process facilitates the formation of spherical particles whereas MA particles are usually smoothed (but are not spherical). Thus, MA powders have higher specific surface and irregular shaped particles that atomized particles of similar radius. Gas atomization favours the formation of the amorphous phase, however, if a high dispersion of particle sizes is produced, depending on the particle size, the structure can be amorphous or a mixture of amorphous and nanocrystalline [18]. With this technique, the contamination from the MA milling tools is avoided. 

Comment: Particle size distribution measurement based on SEM observation is not the most appropriate method. Laser diffraction methods or sieve screening are more suitable and relevant methods. However, I suggest calculating the median particle size as well as the value of the particle size of the first and last 10 %.

Answer: We calculate the median particle size as well as the value of the particle size of the first and last 10 %. Regarding the alternatives, we add a sentence.

The distribution does not correspond to a typical distribution function, either symmetric or asymmetric. Therefore, to compare both distributions, it was decided to perform the calculation of the median particle size as well as the values of the particle size of the first and last 10%. The calculated values were as follows: the first 10% particles size were 8.1 μm and 4.8 μm, the median particle sizes were 16.9 μm and 21.6 μm and the last 10% particles sizes 30.7 μm and 40.6 μm, respectively. As observable, the corresponding values for the distribution of sample B were higher than tohose for sample A.

Alternative methods for determining particle size distribution are sieve screening or laser diffraction laser. In this study, measurements made by sieving (three sieves and therefore four values) are consistent with those determined from microscopic observation.

Comment: In Fig.4. and Fig.5., I suggest adding a peaks description to increase readability.

Answer: We modify both figures taking into account the comment of the referee. The new figures are given in the revised version of the manuscript.

Comment: What is the accuracy of the thermogravimetric method of Curie temperature measurement?

Answer: We add the accuracy (1 K) and also a paragraph to remark the general qualitative accuracy regarding the magnetization of the samples

The modification made to the thermogravimeter would also allow to determine the magnetization of the sample, although for this it would be necessary to fix the external magnet always in the same position and perform a calibration with several magnetic standards (previously analysed with another magnetic measurement technique, such as vibrating sample magnetometry). Consequently, it is better to determine the magnetization of saturation in magnetic devices (such as a vibrating sample magnetometer). However, in consecutive measurements it is always possible to detect in which samples the change in magnetization is greater after the transformation associated to the Curie temperature. To make the comparison, it is necessary to normalize taking into account the mass of each sample used in the experiment. In our case it is detected that the jump is slightly lower in sample A, indicating a somewhat higher initial magnetization. This result must be confirmed with a magnetic measurement (e.g. by magnetic hysteresis cycles).

Reviewer 3 Report

The Authors investigated starting microstructure, thermal stability and magnetic properties of the alloys of compositions Fe-Zr-B-Cu with the difference in B contents, prepared by mechanical alloying from powders. After one time used for milling the investigations confirmed that the increasing B addition lead to the structure rafination decreasing nano - grains, with the dominant bcc crystalline phase. Thermal analysis methods were used to determine following thermal effects and to compare the activation energy determined from the position of the exothermic peak registered probably at the limits of the available temperature range used by the DSC. The presented DSC curves are of a low resolution. Using the mentioned DSC, the Authors should add the base lines for comparison and possibly show the results for the different heating rates to conformed all effects. The activation energy determined from the peak position at the different heating rates was not constant and in the typical range of 200-300 kJ/mol, common for the different processes. The MTGA was further used to determine Curie temperature and the magnetometer was used to determine magnetization and saturation.

Generally, the paper is rather poor and not well explains why this particular compositions were studied. The role of B addition is well known. The main peak in the DSC diagrams was interpreted as effect of crystallization, what is doubtful, as the alloys after MA are crystalline (proved by the XRD). The mechanism of the thermally induced  coalescence and growth of the nano - grained alloys is different from heterogeneous crystallization from melt or transition from the glassy metallic state to the crystalline one. In a results the meaning of the main thermal effect  should be precisely explained to define what concerns the activation energy. Also the Tab. 1 do not supply comparison for the value of the activation energies, as values concerning amorphous alloys results from another type of processes proceeding in heating. Finally, English grammar should be carefully controlled.

The Authors investigated starting microstructure, thermal stability and magnetic properties of the alloys of compositions Fe-Zr-B-Cu with the difference in B contents, prepared by mechanical alloying from powders. After one time used for milling the investigations confirmed that the increasing B addition lead to the structure rafination decreasing nano - grains, with the dominant bcc crystalline phase. Thermal analysis methods were used to determine following thermal effects and to compare the activation energy determined from the position of the exothermic peak registered probably at the limits of the available temperature range used by the DSC. The presented DSC curves are of a low resolution. Using the mentioned DSC, the Authors should add the base lines for comparison and possibly show the results for the different heating rates to conformed all effects. The activation energy determined from the peak position at the different heating rates was not constant and in the typical range of 200-300 kJ/mol, common for the different processes. The MTGA was further used to determine Curie temperature and the magnetometer was used to determine magnetization and saturation.

Generally, the paper is rather poor and not well explains why this particular compositions were studied. The role of B addition is well known. The main peak in the DSC diagrams was interpreted as effect of crystallization, what is doubtful, as the alloys after MA are crystalline (proved by the XRD). The mechanism of the thermally induced  coalescence and growth of the nano - grained alloys is different from heterogeneous crystallization from melt or transition from the glassy metallic state to the crystalline one. In a results the meaning of the main thermal effect  should be precisely explained to define what concerns the activation energy. Also the Tab. 1 do not supply comparison for the value of the activation energies, as values concerning amorphous alloys results from another type of processes proceeding in heating. Finally, English grammar should be carefully controlled.

Many small faults in English.  The terms  mechanical alloying, milling, powder technology are used in the same sense. Please, use one term in your Abstract and manuscript. The best is MA. 

Author Response

Comments and Suggestions for Authors

Comment: The Authors investigated starting microstructure, thermal stability and magnetic properties of the alloys of compositions Fe-Zr-B-Cu with the difference in B contents, prepared by mechanical alloying from powders. After one time used for milling the investigations confirmed that the increasing B addition lead to the structure rafination decreasing nano - grains, with the dominant bcc crystalline phase. Thermal analysis methods were used to determine following thermal effects and to compare the activation energy determined from the position of the exothermic peak registered probably at the limits of the available temperature range used by the DSC. The presented DSC curves are of a low resolution. Using the mentioned DSC, the Authors should add the base lines for comparison and possibly show the results for the different heating rates to conformed all effects. The activation energy determined from the peak position at the different heating rates was not constant and in the typical range of 200-300 kJ/mol, common for the different processes. The MTGA was further used to determine Curie temperature and the magnetometer was used to determine magnetization and saturation.

Answer: We agree the comments of the referee.

Regarding the DSC scans, we add information about the processes involved in each temperature interval and also the baselines for comparison as remarked by the referee. The new figure is in the revised version of the manuscript.

Regarding the activation energy values versus the transformed fraction a comment is added. Furthermore, we improve the discussion about DSC analysis to determine the activation energy taking into account the comments of this referee and one of the other two referees, new references were also added.

It should be remarked that as lower the Fe/B ratio, the main crystallization peak is shifted to higher temperatures (about 20 K). Thus, partial substitution of Fe by B increases the thermal stability of the original nanocrystalline phase produced in the mechanical alloying process.

Thus, the activation energy is that corresponding to the crystal growth mechanism. In nanocrystalline alloys with similar composition, the same phenomena was detected [25-34]. At high transformed fractions, the degree of transformation slows down and the local activation energy decreases [44], probably due to a higher influence of the diffusion as main mechanism. There is probably an impingement between the crystal grains. One of the aspects to highlight is that the isoconversional method here applied is not associated with any kinetic model. Therefore, it allows to parameterize the evolution of the activation energy as a function of the transformed fraction neglecting the effects of a complex and variable kinetics during the transformation (nucleation, growth, impingement). On the other hand, linear methods are based on concrete hypotheses. For example, the Kissinger equation (used in the analysis of the Figure 6) was obtained assuming: a) that the fraction transformed at the peak temperature is the same in all experiments regardless of the heating rate and b) the transformation rate (crystal growth in our case) is maximum at the peak temperature.

With respect to the thermal stability of both samples after 50 hours of grinding, sample B (which has a higher boron content and therefore a lower Fe/B ratio) is the one with the best thermal stability against the main crystallization process. In general, greater thermal stability is associated with two aspects: a) a higher transformation temperature and b) a higher activation energy. In this study both parameters are better in alloy B. From a technological point of view, what is desirable is a greater thermal stability of the nanocrystalline phase, since crystal growth causes an increase in coercivity and the magnetically soft response.

The addition of Cu facilitates the reduction of a) the crystallization temperature and b) the activation energy [36]. In Fe based alloys with a minor addition of Cu, values ranging from 177 to 233 kJ/mol were calculated [37]. Thus, the identification of the process as crystallization or as crystalline growth is usually done taking into account the room temperature nanocrystalline state [26, 34]. Likewise, the formation of borides is found at higher temperatures if compared with those of bcc Fe rich crystallization [38, 39].

Whit respect to the magnetic TG curves analysis we also agree the referee comment. We add a discussion about it. Likewise, we use magnetic hysteretic cycles to check the coherence in the tendency and introduce an additional paragraph.

The modification made to the thermogravimeter would also allow to determine the magnetization of the sample, although for this it would be necessary to fix the external magnet always in the same position and perform a calibration with several magnetic standards (previously analysed with another magnetic measurement technique, such as vibrating sample magnetometry). Consequently, it is better to determine the magnetization of saturation in magnetic devices (such as a vibrating sample magnetometer). However, in consecutive measurements it is always possible to detect in which samples the change in magnetization is greater after the transformation associated to the Curie temperature. To make the comparison it is necessary to normalize taking into account the mass of each sample used in the experiment. In our case it is detected that the jump is slightly lower in sample A, indicating a somewhat higher initial magnetization. This result must be confirmed with a magnetic measurement (e.g. by magnetic hysteresis cycles).

Thus, the sample with lower nanocrystalline size has lower coercivity.

It is found that the magnetization of saturation has a slightly higher value in sample A. This result is consistent with that detected by magnetic thermogravimetry (Figure 8) where sample A shows a slightly higher variation than that of sample B at the Curie temperature (transition from ferromagnetism to paramagnetism).

Comment: Generally, the paper is rather poor and not well explains why this particular compositions were studied. The role of B addition is well known. The main peak in the DSC diagrams was interpreted as effect of crystallization, what is doubtful, as the alloys after MA are crystalline (proved by the XRD). The mechanism of the thermally induced  coalescence and growth of the nano - grained alloys is different from heterogeneous crystallization from melt or transition from the glassy metallic state to the crystalline one. In a results the meaning of the main thermal effect  should be precisely explained to define what concerns the activation energy. Also the Tab. 1 do not supply comparison for the value of the activation energies, as values concerning amorphous alloys results from another type of processes proceeding in heating. Finally, English grammar should be carefully controlled.

Answer:

We add information about the composition selection in the introduction section.

The main difference between the two Fe-Zr-B-Cu alloys produced is the relative content of Fe and B. In one the ratio is 85/8 and in the other 80/13. As iron is the element that contributes magnetism to the alloy, it is expected that the magnetization of the sample with a ratio of 85/8 is greater than that of the ratio of 80/13. However, it is known that the reduction of the nanocrystalline size favors the magnetic response (usually by decreasing coercivity) and that the addition of boron in the Fe base alloys favors either the formation of an amorphous phase or the formation of a nanocrystalline phase with smaller nanocrystals. Likewise, the amorphous induced phase formation is usually associated to a loss of the magnetization of saturation [23], which is an undesired effect for soft magnetic behavior. It is therefore a question of ascertaining in two specific compositions whether one effect can counteract, at least partially, the other or not. Also, to verify the thermal stability of both alloys, since the crystalline growth (provoked on heating the samples) usually entails the loss of soft magnetic behavior. The percentage of copper and zirconium is the same in both alloys, therefore its effect on the different thermal and thermomagnetic response, or in the microstructure can be considered less than that due to the relative content of Fe and Nb. Looking at its role, the Nb being an element of high atomic size will normally be located on the neighboring of the crystalline grains hindering the crystalline growth whereas the well dispersed copper atoms favor the formation of multiple nanocrystals and higher density of nanocrystals also difficult the formation of large sized crystals.

Comments on the Quality of English Language

Comment:

Many small faults in English.  The terms  mechanical alloying, milling, powder technology are used in the same sense. Please, use one term in your Abstract and manuscript. The best is MA. 

Answer:

The English has been checked by an English native speaker.

We agree the comment and modify. Mechanical alloying is labelled as MA the first time introduced, milling has been replaced by MA, and powder technology is only written once.

Round 2

Reviewer 1 Report

My conerns have been addressed in the revised manuscript. 

Author Response

We agree the comment of this reviewer.

"My conerns have been addressed in the revised manuscript. "

Reviewer 3 Report

After the last corrections the paper is clear and English also good. In my opinion your paper is valuable and especially I appreciate your presentation of the applicability of the thermal analysis in the investigation of the magnetic alloys. Congratulations.

Author Response

We agree the comments of the reviewer.

"After the last corrections the paper is clear and English also good. In my opinion your paper is valuable and especially I appreciate your presentation of the applicability of the thermal analysis in the investigation of the magnetic alloys. Congratulations."